# The Interplay of Socioecological Determinants of Work–Life Balance, Subjective Wellbeing and Employee Wellbeing

**DOI:** 10.3390/ijerph18094525

**Published:** 2021-04-24

**Authors:** Ka Po Wong, Fion Choi Hung Lee, Pei-Lee Teh, Alan Hoi Shou Chan

**Affiliations:** 1Department of Systems Engineering and Engineering Management, City University of Hong Kong, Kowloon Tong, Hong Kong, China; alan.chan@cityu.edu.hk; 2Faculty of Business, UOW College Hong Kong, Kowloon, Hong Kong, China; fhung@uow.edu.au; 3Gerontechnology Laboratory, School of Business, Monash University Malaysia, Bandar Sunway 47500, Malaysia; teh.pei.lee@monash.edu

**Keywords:** work–life balance, employee wellbeing, subjective wellbeing, quality time allocation, socioecological determinants

## Abstract

Today’s workers are struggling to achieve a balance between their work and personal life roles because of both specific needs and limited resources. This study explored the socioecological factors that influence work–life balance (WLB) and how they operate. The relationships between WLB, subjective wellbeing, employee wellbeing and quality time allocation were examined. A total of 1063 responses were received, using an online survey. The results show that relational, community and societal factors directly influenced the individual factors and were indirectly associated with perceived WLB. Individual factors (i.e., personal feelings, behaviours and health) were found to be the crucial determinants of an individual’s perceived WLB. It was found that WLB positively correlated with employee wellbeing and quality and quantity of personal life-time. Subjective wellbeing was found to be a significant moderator in the relationship between WLB and its outcomes. This study demonstrated the process of how workers determine their own WLB by applying the socioecological framework for categorising the determinants and suggested new avenues that improve the whole wellbeing of workers and also foster long-term development of organisations.

## 1. Introduction

Balancing between work and personal life is a daunting challenge for workers today because of technological advancement, demographic change, influences of social norms and changes in individual desires. In Hong Kong, many workers deal with work-related calls and emails outside of office hours, thanks to highly accessible technologies that provide a convenient means of communication [1]. Extending time at the office has become a common norm in most workplaces because some workers believe that working overtime can fulfil the market demand and enhance their own prospects [2]. To mitigate the problem of long working hours, the Hong Kong government has promoted family-friendly policies, but these are non-mandatory, which has reduced their effectiveness [3,4]. Working almost continually, as is facilitated by the advanced technologies and also by social demands, seems to reduce the resources and energies of an individual in their family, social life and private time, which creates tension in the relationship with family and friends [5] and leads to high pressure and exhaustion [6]. In the long run, poor health may result in low productivity and increased absenteeism. As a result of these trends, work–life balance (WLB)—maintaining healthy personal and organisational growth—has become a critical element for both workers and employers. The present study focused on the exploration of determinants and outcomes of WLB and considered the determinants as social needs that influence workers to strive towards WLB. This study aimed to understand the interplay between these factors and WLB beyond the typical focus (e.g., the direct influencing power of factors on WLB) and move the research onwards by testing more complex relationships regarding WLB.

### 1.1. Literature Review

Many scholars have suggested theoretical approaches to define and describe WLB. Initially, the focus amongst researchers was simply the balance between work and family. Greenhaus et al. [7] adopted the role engagement approach to define WLB, which refers to equality in time, involvement and satisfaction in work and family roles. Frone [8] suggested the minimal conflict approach to describe WLB, which means a lack of conflict between work and family life. In the role satisfaction approach, WLB refers to balance satisfaction among life roles or expectations [9,10]. In 2013, Haar [11] critiqued a simplistic approach to work–family balance as insufficient to address the whole life of an individual and suggested broadening the scope of ‘family’ to the whole personal life. WLB has emerged as a concept both in wider society and academic areas and become a universal term. Importantly, single workers, married workers and working parents were all benefitted from WLB. Although some research employed a low level of conflict between work and life or the counter-off between conflict and enrichment to represent WLB, research has demonstrated that WLB is distinct from work–family conflict or enrichment and that WLB mediates work–family conflict or enrichment [11,12].

The International Labour Organisation has formulated a guide to developing balanced working time, to assist in achieving WLB [13]. In the ILO work, working hours seem to be an important factor impacting WLB, whereas other researchers have identified various antecedents influencing WLB [14,15,16,17]. These authors found working hours, technologies, work autonomy, work demands and family support to be positively associated with WLB. The level of WLB might not be directly influenced by these determinants. For example, workaholism—workaholic people enjoy their work and working long hours might not have a negative impact on their WLB. Little is known about the influencing powers between the determinants of WLB. Haar and Brougham [16] indicated that many studies have failed to test more complex relationships in WLB, and the testing of the antecedents of WLB was limited. However, these authors also found that the factors (e.g., working hours, work demands and support) influenced WLB—which in turn impacted job satisfaction and organisational commitment.

WLB seems to have several positive outcomes for organisations and individuals. For organisations, WLB can improve an individual’s work performance, productivity and relationship with co-workers. However, few studies have examined the relationship between individual-level WLB and employee wellbeing, as previous research focused on the effects of organisational-level WLB on employee wellbeing (an exception being Zheng et al. [18]). Regarding the positive effects for the individual, however, WLB is beneficial to the involvement in family activities, social life and leisure time in which the amount of quality time spent on these activities is increased. Veal [19] discussed the necessity for leisure (including rest, entertainment and family time) and the relationship between work and leisure. Yet, to our knowledge, no empirical study has yet evaluated the relationship between WLB and leisure, and hence further investigation of the positive effects of WLB on the involvement in personal time and leisure is needed. Gröpel and Kuhl [20] found that WLB was positively related to subjective wellbeing. Peiró et al. [21] found that workers with high levels of happiness were more productive than those with a low level of happiness. Oishi et al. [22] showed that individual differences in subjective wellbeing affected both work and non-work situations. That is, an individual who has a high level of life satisfaction is likely to experience positive effects in other situations. This finding implies that subjective wellbeing will have a positive impact on work performance. The moderating role of subjective wellbeing on the relationship between WLB and its outcomes has been neglected in the past studies, and thus the moderating effect of subjective wellbeing on such relationships requires further assessment.

This study addressed these issues and made several significant contributions. First, we responded to the call from Haar and Brougham [16] for testing a more complex-mediated relationship of WLB by using a socioecological framework to categorise the antecedents. According to ecological systems theory, the socioecological framework assumes that the changes in personal outcomes are not only affected by individual-level factors such as ability and feelings but also through the interactions with social, cultural, economic factors and contexts where the person lives [23]. Employing this framework to examine the dynamic interplay between environmental and individual factors regarding WLB can lead to a better understanding of their influences on WLB. Second, we examined an unexplored empirical relationship between socioecological factors and self-perceived WLB by testing Leslie et al.’s [24] suggestion that socioecological context is the forerunner of work-life ideology. Previous studies have often postulated environmental factors (e.g., workplace policies, job characteristics and relationships with others) as the direct determinants of self-perceived WLB (e.g., [25,26,27]). Third, we articulated a new area of inquiry for needs theory. Needs theory is an extension of Maslow’s hierarchy theory [28] in that high-level needs are summarised as the social needs for achievement, power and affiliation. In the developing model, the relationships with others, workplace policies, job characteristics and societal influences were theorised as social needs [29]. McClelland [30] highlighted the importance of fulfilment of social needs in order to motivate work performance and improve the attitudes of workers. Our study hypothesised that fulfilling social needs might benefit the roles in both work and non-work domains. Fourth, multiple discrepancies theory was applied to draw a connection between self-perceived WLB and subjective wellbeing. Multiple discrepancies theory suggests that happiness and satisfaction are the outcomes of the perceived gap between what one actually has and what one expects to have compared with others or the past [31]. How an individual is conscious of WLB through gauging subjective wellbeing is highlighted. Lastly, the study considered the moderating effect of subjective wellbeing on the association of self-perceived WLB with employee wellbeing and quality and quantity of personal life-time. This study drew attention to the effects contributed by individual-level WLB rather than organisational-level WLB. Hence, the impacts of self-perceived WLB on health state at work and quality time spent on leisure, social life and family might be moderated by subjective wellbeing, which may reduce the effects. Overall, useful insights into the importance of the influencing power of external and internal factors influencing WLB were provided by this study, and empirical evidence demonstrating the importance of WLB for individuals and organisations was shown.

### 1.2. Theoretical Background and Hypotheses

#### 1.2.1. Needs Theory, Socioecological Determinants and Work–Life Balance

Needs theory has been widely applied in the literature of management and organisational behaviour and has explained the effects of the needs for achievement, power and affiliation on the attitudes and behaviours of an individual. McClelland [30] emphasised that most human needs are not physiological but social and showed that human social needs are not innate but acquired and derived from the environment, experience, training and education. Satisfaction with all life domains should be attained to maintain a healthy WLB. However, each individual’s needs are different, and the socioecological approach considers the complex interaction amongst individual, relational, community and societal factors. The range of external determinants influencing the perceived WLB could be understood through the socioecological model. By applying the needs theory to the socioecological approach, McClelland [30] proposed three types of needs, including needs for achievement, power and affiliation, which were theorised as the external determinants in this study. These needs must all be satisfied to maintain a healthy WLB. Scholarly exploration of the impacts of social needs on intra-individual variables that directly influence self-perceived WLB has been limited (e.g., [25,32,33]). To address this gap, we adopted a socioecological approach to depict the relationship between social needs and WLB, in which the determinants were separated into two levels. The relational, community and societal factors, which were defined as social needs, were the first-level indirect influencing socioecological determinants in our model. Relational factors included satisfaction with supervisor, satisfaction with colleagues, family relationships and peer influences. Community and societal factors included workplace policies, job characteristics and societal influences. The second-level influencing factors were the intrapersonal-level factors, which are affected by the indirect influencing factors and are directly related to WLB. Intrapersonal-level factors included feelings at work, personal time use and health condition. The environmental factors would primarily affect the feelings and consciousness of an individual.

##### Relational Factors and Intrapersonal-Level Factors

It has been suggested that building positive and healthy relationships with others may improve the balance between work and non-work roles because it contributes to the physical and mental wellbeing of workers [34]. A healthy relationship implies the perception of understanding and support between the parties [35]—open communication, trust, respect and feeling comfortable are qualities of the relationship. Using the concept of needs theory, if a worker feels loved and a sense of belonging to social groups, including families, companies and friendship groups, his/her wellbeing is improved [36]. Based on the role satisfaction approach, workers feeling satisfied with the relationships and behaviours of their supervisors, colleagues, family members and friends may have better general wellbeing. We, therefore, postulated that workers who have more understanding attitudes and support from supervisors, colleagues, families and peers will have better physical and mental health and time management ability.

**Hypothesis** **1.**
*Relational factors are positively related to intrapersonal-level factors.*


**Hypothesis** **2.**
*(2a) Satisfaction with supervisor, (2b) satisfaction with colleagues, (2c) family relationship and (2d) peer influences are positively related to relational factors.*


##### Community and Societal Factors and Intrapersonal-Level Factors

Favourable community and societal factors are at least as important for workers’ improving physical and mental health and time management ability [37,38]. Support from government and organisations (e.g., the implementation of employee-friendly policies, provision of family-friendly facilities and improvement of welfare subsidies) enhances the wellness of workers as well as their attitudes towards the companies [39]. Job autonomy, meaningfulness and interest are positively associated with happiness and enhanced time planning [37,40]. Further, prolonged use of communication technologies and the social norm of long working hours are detrimental to the working attitudes and wellness of workers [41,42]. Therefore, the following hypotheses were proposed.

**Hypothesis** **3.**
*Community and societal factors are positively related to intrapersonal-level factors.*


**Hypothesis** **4.**
*(4a) Workplace policies, (4b) job characteristics and (4c) societal influences are positively related to community and societal factors.*


##### Intrapersonal-Level Factors and Work–Life Balance

Intrapersonal-level factors commonly include the attitudes, experiences, knowledge and skills of an individual [43] that are related to the internal information process [44]. Environmental factors directly influence the feelings, emotions and behaviours that impact the perceived WLB. Positive surrounding circumstances, for example, support from supervisors, good family relationships and favourable workplace policies, positively affect the attitudes, moods and health of workers [44]. Workers with positive attitudes and health conditions are likely to perceive a better balance in their lives. In contrast, workers who perform ineffectively, have a mood disorder and feel a lack of social connectedness may perceive an imbalance. Thus, we presupposed that the individual’s perceived feelings and health condition predominantly influence WLB.

**Hypothesis** **5.**
*Intrapersonal-level factors are positively related to WLB.*


**Hypothesis** **6.**
*(6a) Feelings at work, (6b) personal time use and (6c) health condition are positively related to intrapersonal-level factors.*


#### 1.2.2. Multiple Discrepancies Theory, WLB and Subjective Wellbeing

Multiple discrepancies theory is generally adopted to explain quality of life, subjective wellbeing, satisfaction and happiness by evaluating the perceived gaps between actuality and various standards. Examples include what one possesses and what one desires to have and what another possesses and what one believes they should have. Subjective wellbeing refers to how an individual evaluates his or her life cognitively [45]. Multiple discrepancies theory explains how subjective wellbeing is affected by perceived balance. That is, if one was satisfied with the work and personal life domains through an evaluation based on self-predetermined standards and expectations or comparing with others’ circumstances, then perceived gaps between actual and ideal conditions might be low, which yields a high level of subjective wellbeing. Based on the needs theory, a high level of balance is likely to be associated with the satisfaction of an individual’s needs. When workers perceived a balance in work and personal life, they also experienced positive subjective wellbeing. Subjective wellbeing focuses on mental states, life satisfaction and physical health. Several studies have shown that WLB is positively correlated with subjective wellbeing [25,46,47]. Therefore, we suggested that the WLB of workers is achieved when the workers perceive virtually no gap between their actual and expected situations, and this results in positive subjective wellbeing.

**Hypothesis** **7.**
*WLB is positively related to subjective wellbeing.*


**Hypothesis** **8.**
*(8a) Personal life satisfaction and (8b) health outcomes are positively related to subjective wellbeing.*


#### 1.2.3. Multiple Discrepancies Theory, WLB and Subjective Wellbeing

Employee wellbeing is the overall satisfaction and happiness of an employee at the workplace [48] and includes the concepts of job satisfaction, job-related mental strain and job-related depression [49]. Stress, burnout and personality traits were identified as significant predictors of job satisfaction and job-related mental health [50,51,52]. Applying needs theory and multiple discrepancies theory, balance is associated with the satisfaction of workers and lack of a perceived gap between expected and actual needs. Workers with a healthy WLB will in turn have better mood, emotions and health, leading to improving attitudes and behaviours towards work and the arrangement of personal affairs. Many studies have focused on investigating the effect of organisational-level WLB and the direct relationship between organisational-level WLB and employee wellbeing [18,53,54]. We focused on the individual—instead of the organisational-level WLB and the moderating effect of subjective wellbeing on the relationship between WLB and employee wellbeing. Moreover, the factors we found for employee wellbeing were work attitudes, work behaviours, career aspirations and cooperation with colleagues. We proposed that subjective wellbeing moderates the relationship between WLB and employee wellbeing. WLB does not impact employee wellbeing directly, owing to the moderating effect of personal emotions and health conditions.

**Hypothesis** **9.**
*Subjective wellbeing moderates the effect of WLB on employee wellbeing.*


**Hypothesis** **10.**
*(10a) Work attitudes, (10b) work behaviours, (10c) career aspirations and (10d) cooperation with colleagues are positively related to employee wellbeing.*


#### 1.2.4. Subjective Wellbeing as a Mediator of the Relationship between WLB and Quality and Quantity of Personal Life-Time

Quality and quantity of personal life-time—which means a sufficiency of time to spend with someone or on something with full attention, without having to cope with other matters—refers in this context to the quality and sufficiency of time able to be spent on family, leisure and social life. Wiese et al. [55] conducted a meta-analysis and reported that a positive association exists between spending leisure time on physical activities and subjective wellbeing. Similarly, Hribernik and Mussap [56] stated that subjective wellbeing is influenced by leisure satisfaction. Thomas et al. [57] argued that the family context is of significance to subjective wellbeing in life. That is, subjective wellbeing positively influences the time spent on family, leisure and social life to some extent. The hypothesis suggests that WLB is likely to affect the subjective wellbeing of an individual, which has been identified as one of the important factors in the quality and quantity of time for family, leisure and social life.

**Hypothesis** **11.**
*Subjective wellbeing moderates the effect of WLB on quality and quantity of personal life-time.*


**Hypothesis** **12.**
*(12a) Family context, (12b) leisure time and (12c) social life are positively related to quality and quantity of personal life-time.*


## 2. Materials and Methods

### 2.1. Measures

The questionnaire was composed of seven main sections: (a) relational factors, (b) community and societal factors, (c) intrapersonal-level factors, (d) work–life balance, (e) subjective wellbeing, (f) employee wellbeing and (g) quality and quantity of personal life-time. Relational factors consisted of satisfaction with supervisor, satisfaction with colleagues, family relationship and peer influences. Then, community and societal factors comprised workplace policies, job characteristics and societal influences. Intrapersonal-level factors included feelings at work, personal time use and health condition. Subjective wellbeing embodied personal life satisfaction and health outcomes. Employee wellbeing consisted of work attitudes, work behaviours, career aspirations and cooperation with colleagues. The quality and quantity of personal life-time included the family context, leisure time and social life. A total of 91 statements regarding the perceived determinants and outcomes of WLB were generated, based on the review of previous studies, and twenty of these were new survey items. The other 71 items were validated by researchers (e.g., [58,59,60]). All items were measured with a seven-point Likert scale which ranged from 1 (completely disagree) to 7 (completely agree). Table 1 shows the contents of each item.

The language used for the questionnaire was Chinese. Four independent bilingual translators, who were not associated with the study, were invited to conduct the translation to ensure the reliability of the translated questionnaire [58]. Two of the translators translated the English version into Chinese, and another translated the Chinese version into English. The discrepancies amongst the translators were identified and discussed with the researchers until all were satisfied with the statements.

#### 2.1.1. Relational Factors of Work–Life Balance

Satisfaction with the supervisor was measured using five items adapted from Chen et al. [59], Chen et al. [60] and Jernigan and Beggs [61]. Satisfaction with colleagues was measured using three newly developed items. The family relationship was measured using three items, two of which were adapted from Fok et al. [62] and Hoffman et al. [63] and one was a newly developed item. Peer influences were measured using three items, one of which was developed by Bukowski et al. [64] and two were newly developed items.

#### 2.1.2. Community and Societal Factors of WLB

Workplace policies were measured using six items adapted from Smith and Gardner [65]. Job characteristics were measured using four items, three of which were adapted from Allen and Meyer [66] and Törnquist Agosti [67] and one was a newly developed item. Societal influences were measured by five items, two of which were adapted from Pelletier et al. [68] and three were newly developed items.

#### 2.1.3. Intrapersonal-Level Factors of Work–Life Balance

Feelings at work were measured using three items, two of which were adapted from Brayfield and Rothe [69] and Joseph et al. [70] and one was a newly developed item. Personal time use was measured using three items, two of which were developed by Wong and Ko [58] and one was a newly developed item. Health condition was measured using six items, five of which were adapted from Bender et al. [71], Lucia-Casademunt et al. [54] and Husin et al. [72], and one was a newly developed item.

#### 2.1.4. Work–Life Balance

WLB was measured using a four-item scale developed by Brough et al. [73].

#### 2.1.5. Subjective Wellbeing

Personal life satisfaction was measured using six items adapted from De Pater et al. [74], Joseph et al. [70], Keyes [75], Macaskill and Taylor [76] and Wong and Ko [58]. Health outcomes were measured using ten items adapted from Cohen et al. [77], Hays et al. [78], Keyes [75], LeBourgeois [79], Milton et al. [80] and Schwartz et al. [81].

#### 2.1.6. Employee Wellbeing

Work attitudes were measured using nine items adapted from De Pater et al. [74], Haider et al. [82], Lawler and Hall [83] and O’Driscoll and Beehr [84]. Work behaviours were measured using six items, four of which were adapted from Haider et al. [82] and Hill et al. [85] and two were newly developed items. Career aspirations were measured using three items adapted from Kraimer et al. [86]. Cooperation with colleagues was measured using three items adapted from Ladd and Henry [87].

#### 2.1.7. Quality and Quantity of Personal Life-Time

The family context was measured using three items, one of which was developed by Vaughn and Baier [88] and two were newly developed items. Leisure time was measured using three newly developed items. Social life was measured using three items developed by Keyes [75] and Wong and Ko [58].

#### 2.1.8. Demographics

Demographic information involved age, gender, educational level, marital status, number of dependents, religious belief, average monthly income, average working hours per week, average working hours per day, employment condition, work status, working age, work position and industries.

### 2.2. Data Collection and Sample Size

The data were collected from an online survey platform, Qualtrics. The procedure of the online survey was approved by the College Research Ethics Sub-committee of the City University of Hong Kong. This survey was a self-report design that was administrated online to gather confidential data from participants in different organisations. The first page of the survey explained the background of the survey and the participant agreement. If the individuals understood and consented to the conditions, they could click ‘Agree’ to start the survey. The inclusion criteria were to be aged eighteen and above and currently employed in Hong Kong. Based on simple random sampling, invitation emails were sent to twenty-five trade unions according to the list from the Hong Kong Confederation of Trade Unions (HKCTU) [89]. HKCTU is an influential labours group in Hong Kong, and several researchers have collected sample data based on the affiliates under this union [90,91]. Finally, three trade unions assisted in distributing the online survey, and nine organisations participated in conducting the survey. The sample comprised the workers of key industries of Hong Kong, namely financial services, tourism, trading and logistics, professional services, producer services, cultural and creative industries, medical services and innovation and technology.

### 2.3. Data Analysis

Exploratory factor analysis (EFA) was adopted to test the preliminary factor analysis of the instruments by using SPSS 24.0. The measurement and structural model were analysed through confirmatory factor analysis (CFA) and structural equation modelling (SEM), respectively, using SPSS 24.0 and AMOS 24.0. The factor loadings in EFA should be greater than 0.4 to meet the standard [92]. Maximum likelihood estimation (MLE) was used for the models of CFA and SEM. To assess the fitness of the measurement and structural model, five goodness of fit indices, namely, chi-square to its degree of freedom (χ^2^/df), Tucker–Lewis Index (TLI), comparative fit index (CFI), standardized root mean squared residual (SRMR) and root mean square error of approximation (RMSEA), were used [93]. Factor loadings, composite reliability (CR) and average variance extracted (AVE) were used for convergent validity. The square root of AVE for the latent factors was larger than the inter-factor correlations indicating the verification of discriminant validity [94].

## 3. Results

### 3.1. Characteristics of the Sample

A total of 1063 valid responses were collected. The demographic information of the respondents is as follows: 52.96% of the respondents were male, and 47.04% were female. A total of 15.52% of the respondents were aged 18–24; 31.14% were aged 25–34; 11.48% were aged 35–44; 24.84% were aged 45–54; 16.27% were aged 55–64; 0.75% were aged greater than 65. Of the respondents, 43.55% had completed a bachelor’s degree or above. Moreover, 44.40% were single, 1.22% were single with children, 7.43% were married without children, 45.34% were married and with children, and 1.60% were divorced/separated/widowed. Of the respondents, 73.66% had no religious belief. The average monthly personal income of 49.95% of the respondents was HKD 22,865 or below; for 43.84%, it was HKD 22,865–70,090; for 4.61%, HKD 70,091–140,560; for 1.41%, HKD 140,560 or above; only two respondents did not provide this information. The average working hours per week of 1.22% of the respondents were 10 to fewer than 20; for 5.27%, they were 20 to fewer than 29; for 9.60%, 30 to fewer than 39; for 44.31%, 40 to less than 49; for 21.35%, 50 to fewer than 59; for 14.02%, 60 to fewer than 69; for 3.10%, 70 to fewer than 79; for 0.94%, 80 to fewer than 89; for 0.19%, 100 to fewer than 109. Moreover, 87.86% were full-time workers, and 12.14% were part-time workers. Of the respondents, 93.51% were employed and 3.76% were self-employed. The participants came from more than fifteen industries.

### 3.2. Preliminary Analyses of the Measurement Model

The newly developed instruments were selected to be tested in the EFA because of the large number of items in the questionnaire. As shown in Table 2, the factor loadings of the 20 new instruments were greater than 0.4, which met the cut-off standard [92]. The results of CFA showed an acceptable overall model fit, except χ^2^/df (χ^2^/df = 10.48, CFI = 0.925, TLI = 0.909, RMSEA = 0.075, SRMR = 0.058; [93]). The value of χ^2^/df represented the relative deviation between empirical data and the model. This metric is sensitive to sample size [95]. Therefore, in the case of a large sample size, although the specification misalignment of the model is small, the model can be rejected [95]. An RMSEA value up to 0.08 represents reasonable errors of approximation in the population [96,97]. Thus, these conclusions are based on fit indices of CFI, TLI, SRMR and RMSEA that are independent of the sample size. Table 3 demonstrates the convergent construct validity of the instrument. The standardised factor loadings of all items were greater than 0.60 [94], the CR was greater than 0.70, and the AVE was greater than 0.50, which satisfied the requirement of convergent validity. Discriminant validity was satisfied as the square root of AVE for the latent factors was larger than the inter-factor correlations (see Table 4). Table 5 shows the Pearson correlations between demographic information and the latent variables.

### 3.3. Test of Hypotheses

To verify the structural model, SEM was used, and the overall model satisfied the criteria of the indices of goodness of fit, except χ^2^/df (χ^2^/df = 9.31, CFI = 0.933, TLI = 0.920, RMSEA = 0.080, SRMR = 0.059; [93]). An RMSEA value up to 0.08 represents reasonable errors of approximation in the population [96,97]. The conclusions are based on fit indices of CFI, TLI, SRMR and RMSEA that are independent of the sample size. Figure 1 depicts the results of the path analysis of the hypotheses in the proposed model.

All hypotheses received support. For Hypothesis 1, relational factors had a statistically significant, positive relationship with intrapersonal-level factors (b = 0.442, *p* < 0.001). For Hypothesis 2, (H2a) satisfaction with supervisor, (H2b) satisfaction with colleagues, (H2c) family relationship and (H2d) peer influences were positively related to relational factors. For Hypothesis 3, community and societal factors were positively correlated with intrapersonal-level factors (b = 0.555, *p* < 0.001). For Hypothesis 4, (H4a) workplace policies, (H4b) job characteristics and (H4c) societal influences were positively related to community and societal factors. For Hypothesis 5, intrapersonal-level factors were positively associated with WLB (b = 0.859, *p* < 0.001). For Hypothesis 6, (H6a) feelings at work, (H6b) personal time use and (H6c) health condition were positively associated with intrapersonal-level factors. For Hypothesis 7, WLB was positively related to subjective wellbeing (b = 0.424, *p* < 0.001). For Hypotheses 8, (H8a) personal life satisfaction and (H8b) health outcomes were positively associated with subjective wellbeing. For Hypotheses 9 and 11, subjective wellbeing moderated the relations between WLB with respect to (H9) employee wellbeing (b = 0.016, *p* < 0.01) and (H11) quality and quantity of personal life-time (b = 0.023, *p* < 0.01). The results of the significant moderating effect of subjective wellbeing on the relationships between WLB and the two outcomes are reported in Table 6. For Hypotheses 10, (H10a) work attitudes, (H10b) work behaviours, (H10c) career aspirations and (H10d) cooperation with colleagues were positively correlated with employee wellbeing. For Hypotheses 12, (H12a) family context, (H12b) leisure time and (H12c) social life were positively correlated with quality and quantity of personal life-time.

## 4. Discussion

The present study was designed to determine what and how socioecological determinants influenced WLB and the moderating effects of subjective wellbeing on the relationship between WLB and its positive outcomes on individuals and organisations. It is interesting to note that favourable relational and community and societal factors have positive influences on intrapersonal factors which in turn improve the perceived WLB. It was also found that WLB has positive impacts on employee wellbeing and quality time allocation for personal life in which subjective wellbeing moderated these relationships. We used needs theory [30] to understand how the fulfilment of the needs from relationship, community and society can improve personal context and in turn boost WLB. The results of the study further support the idea of Leslie [24] that family, organisation, community and society shape and make the salient features of the relations between work and life. The role of intrapersonal-level factors was found as the predominant factor affecting WLB. A possible explanation for this could be that if an individual’s attitudes and conditions are healthy and positive without being influenced by unfavourable external events, the WLB of the individual should stay at a healthy level. It also suggests that external influences (e.g., relationship with others, job characteristics and social norm of long working hours) might not affect WLB directly and may possibly have an impact on an individual’s thoughts and feelings first. However, such results have not previously been described as studies showed that external events had a direct impact on WLB [16,58,98]. Furthermore, the results show that WLB is beneficial to subjective wellbeing, employee wellbeing and quality and quantity of personal life-time. The findings corroborate studies that have found that having a healthy WLB is beneficial to wellbeing, job and life satisfaction, physical and mental health and organisational commitment [16,99,100]. This study provided evidence for the moderating effect of subjective wellbeing on the relationship between WLB and its positive effects on employee wellbeing and quality time allocation for personal activity. Nevertheless, the investigation on the moderating effect of subjective wellbeing is limited in previous research. Further research is needed to investigate the predominating influencing power of intrapersonal-level factors on WLB and the moderating effect of subjective wellbeing on the relationship between WLB and its positive outcomes.

### 4.1. Contributions to WLB Research

In a nutshell, this study advances the WLB literature by offering a profound perspective of the crucial role of innermost sensation in understanding how individuals perceive WLB while experiencing different external events; and by expanding the scope of needs theory and multiple discrepancies theory in unravelling the formation of balance amongst multiple roles and providing empirical evidence of the linkage of the critical role of subjective wellbeing with WLB. Furthermore, we suggested a novel approach, socioecological systems, in studying the determinants of work–life balance that identifies the dynamic interaction between environmental and individual factors. Using socioecological systems to categorise determinants for this study comes from Maslow’s hierarchy of needs [28]—that the basic needs must first be met to further motivate an individual to attain the higher level of needs. Although prior WLB research has probed the relationship between innermost feelings and WLB [101,102], the significance of the innermost feelings to the interaction with external events and perceived WLB has not previously been considered. Innermost feelings were also the centre of attention in this empirical study since intrapersonal-level factors were the direct predictor of WLB. There are few extant studies that have adopted needs theory and multiple discrepancies theory to elaborate the processes of how social needs affect WLB and how the perceived difference between ideal and reality determines the level of balance. This study, however, illustrated the configuration of WLB by these two theories. The close relationship between subjective wellbeing and WLB has not yet been explored, but we proposed a model to assess how subjective wellbeing responded to WLB and how it impacts employee wellbeing and quality personal time allocation. We contributed several particular contributions to WLB research by advancing the understanding of direct and indirect antecedents impacting WLB, the configuration of perceived balance and the interlinkage between subjective wellbeing and WLB.

Firstly, we affirmed the utility of the socioecological framework in analysing the direct and indirect determinants of self-perceived WLB suggested by Leslie et al. [24]. The results broaden the understanding of scholars as to the dynamic interplay between external and individual factors and reveal how socioecological factors exert influences on self-perceived WLB. A few prior researchers have considered the effects of indirect external factors and direct individual factors on the work–life interface (e.g., [26,103,104]). The results indicate that the most important determinants of self-perceived WLB were innermost feelings, physical functioning and mental health, all of which had a direct influence on self-perceived WLB. However, to date, research had tended to neglect the salient and unique impact of intrapersonal factors on self-perceived WLB. By employing socioecological frameworks to investigate the factors of WLB in this study, we separated the factors into two levels that highlight the extent of the influencing power of different potential factors over WLB.

Secondly, by applying needs theory to the model, we provided evidence with respect to the importance of the indirect determinant, that is, satisfaction of social needs, in predicting self-perceived WLB. The model proposed in this study expanded the scope of inquiry for needs theory [30] by examining the results of meeting these needs beyond the sole impacts on work performance and the attitudes of workers. McClelland [30] suggested that satisfying the need for achievement, affiliation and power can influence how workers set goals, collaborate with other co-workers and motivate teammates. McClelland made no attempt to explain how meeting these social needs benefits the non-work role. This study supports our speculation that satisfying social needs would generate positive outcomes in either work or life domains. Our model extends the scope of the benefits claimed by McClelland [30] and links the entire wellbeing of workers. We emphasised the importance of satisfying social needs in both work and non-work roles to enhance the sustainability of organisational development and community [105,106]. Further, the needs theory of McClelland [30] focused the motivation of workers on achieving organisational goals. This claim, however, seems to overlook the pitfall of long-term engagement in the work role, which uses physical and mental energy and which results in deteriorating performance in both work and non-work roles and deteriorating health conditions [107]. Our study highlighted the prominence of satisfaction of social needs in the overall development and growth of an individual which appears to be linked to the general wellbeing of individuals.

Thirdly, supporting multiple discrepancies theory, our findings suggest that the level of WLB could be determined based on the perceived gap between desire and reality, predicting subjective wellbeing. Although WLB is always an abstract concept amongst most workers, our study discovered that subjective wellbeing would be a suitable mediator to reflect the level of self-perceived WLB as subjective wellbeing reflects a spontaneous and direct response to different events. These results provide a way to measure self-perceived WLB, given the continuous changing of society and personal needs. It is easy for people to share their thoughts, ideas and information through the virtual community, and such sharing may influence the expectations of others. Some prior research has explored the meaning of WLB by using the concept of self-perceived discrepancies, and this study contributed a theoretical explanation for self-perceived WLB.

Fourthly, subjective wellbeing was identified by this study as the potential moderator in the relationship between WLB with respect to employee wellbeing and the quality and quantity of personal time use. These findings reveal that subjective wellbeing may enhance or diminish the impacts of WLB on employee wellbeing and the quality and quantity of personal time use. The measures of subjective wellbeing in this study focused on personal life satisfaction and health outcomes, which were found to be important moderators for WLB in affecting other domains. Most studies investigated the direct relationship between WLB and different outcomes (e.g., [26,108]); subjective wellbeing, however, is the deciding factor that has previously been overlooked. Importantly, the results demonstrate that subjective wellbeing serves as a significant moderating mediator through which WLB affects employee wellbeing and the quality and quantity of personal time use.

Lastly, the existence of ripple effects exerted influence on the model proposed for self-perceived WLB. The results demonstrate that WLB can be affected by a series of unexpected events. This result indicated that a change in environment might ripple out and the mood, feelings and thoughts of an individual would be affected. The rippling process would continue and influence perceived WLB as well as wellbeing and time planning. That is, the deciding factors of self-perceived WLB were subject to the feelings, thoughts and health condition of an individual, who may be sensitised to any elements in the environment. This finding suggests that the objective of training should be an optimistic outlook, in order to achieve a healthy WLB, and a healthy body must be maintained for an optimal daily life.

### 4.2. Practical Implications

This study responded to the question of how to evaluate self-perceived WLB and its positive outcomes by separating the factors involved into two different but linking levels. The results of our research show that satisfying social needs in relationships, community and society was significantly associated with WLB. Further, either employee wellbeing or quality and quantity of personal time use had a positive relationship with WLB by moderating subjective wellbeing. This result sheds light upon the significance of innermost feelings, physical functioning and mental health as the overwhelming influences on self-perceived WLB. In this study, several social needs were identified: relatedness with supervisor and colleagues, workplace policies, job autonomy and interestingness of job tasks. In other words, organisations can formulate tailor-made workplace policies to fulfil the feasible and reasonable requirements of workers so that workers can flexibly manage the work–life interface [109,110,111,112]. One notable factor of significance is the attitudes and behaviours of supervisors towards workers. It is crucial that supervisors review whether or not they assign appropriate workloads to workers, regularly recognise outstanding performance amongst workers, whether they are effective communicators and if they provide suitable resources, since all of these are crucial to enhancing the WLB of workers [113]. Another issue is the use of social media. The high accessibility of social media induces unavoidable connection with others, and comparison with others is inevitable. However, workers are recommended to elevate their spirits rather than pursuing tangible rewards to minimise the tangled dilemma of disparity.

### 4.3. Limitations and Future Research Direction

This study inevitably has some limitations. Firstly, the sample was conducted in only one region, and thus, the results may not be applicable to other countries with distinct politics, economy and cultures. The model suggested in the present study has some reference value to address most work–life interface issues. Secondly, the sample was from particular types of industries and family backgrounds, which have various work characteristics and cultures (e.g., working hours and environment). The different backgrounds of workers may affect the study variables, although some prior studies also adopted the same approach [11,114]. Thirdly, a self-reported approach was adopted in this WLB measure. To resolve this concern, using medical health examination, supervisor appraisal of work performance and family member appraisal of family involvement in the data collection would assist in avoiding subjective evaluation.

This research highlighted some potential research directions. Firstly, the results show that subjective wellbeing provided a significant moderating mediation in the relationship between WLB and its outcomes. In this study, the measures of subjective wellbeing constituted personal life satisfaction and physical and mental health. However, the scope of subjective wellbeing can perhaps be expanded to optimise the measurement of subjective wellbeing. Drawing on multiple discrepancies theory, a more comprehensive measure of WLB can be created constructively for an individual to evaluate their own WLB. Secondly, further study could examine more consequences with WLB by using the moderating effect of subjective wellbeing to consider, for example, financial wellbeing, creativity and skill enhancement [115]. Further, the disrupting occurrence of the COVID-19 pandemic shifted the traditional work mode from office to remote work [116]. This practice may break the accustomed concept that work and personal life are separated into two distinct domains. The virtual work setting is unavoidably brought into our personal lives. The video conferencing call allows workers to be more empathetic to the daily needs of co-workers because workers may understand each other’s family situations. These daily needs have generally been neglected in the past. More importantly, how the change will affect WLB under these circumstances is a vital question that needs further investigation. In addition, given the predominant role of intrapersonal factors on WLB and wellbeing, the impact of personality (e.g., pessimistic, optimistic and envious) on wellbeing can be assessed and different personality types on wellbeing can be identified. The ultimate future direction is to investigate the antecedents of self-perceived WLB by adopting a socioecological framework. This study validated the effectiveness of the socioecological framework in classifying the antecedents into two linking groups, namely, environmental and intrapersonal-level factors. The socioecological antecedents are suggested to be categorised into more levels as it can profoundly help to understand the phenomenon by recognising how an individual respond to external events, which can themselves be categorised into interpersonal, organisational, community-level and public-policy-related factors.

## 5. Conclusions

This study contributes to the HRM and organisational psychology literature in various ways. Firstly, the present study extends our knowledge of the significant ripple effects of socioecological factors on self-perceived WLB, linking employee wellbeing and time allocation for personal activities. Secondly, socioecological factors were identified as social needs, which explained how a healthy WLB was able to be achieved when these needs were satisfied. Thirdly, the results show that the perceived gaps in meeting the social needs between expectation and actuality were associated with self-perceived WLB. Lastly, our study indicated the significant moderating effect of subjective wellbeing on WLB. Thus, this research is pertinent to either HRM scholars or practitioners interested in enhancing workers’ WLB to promote employee wellbeing and quality of life.

## Figures and Tables

**Figure 1 ijerph-18-04525-f001:**
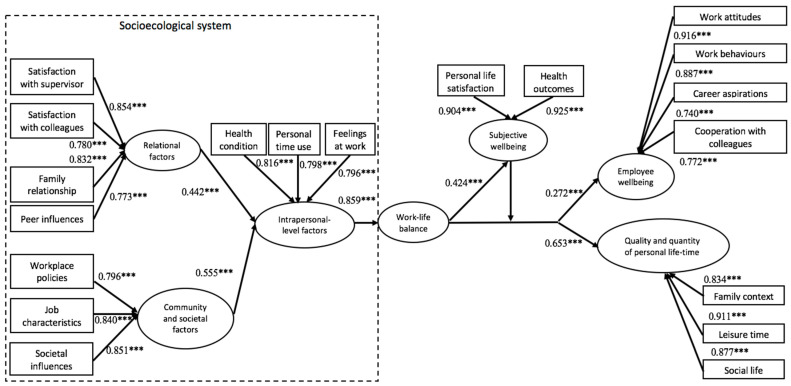
Result of the structural model with standardised estimates where *p* ≤ 0.001. *** *p* ≤ 0.001.

**Table 1 ijerph-18-04525-t001:** Contents of each item.

Socioecological Factors	Items	Content
Satisfaction with Supervisor (SS)	SS1	My supervisor assigns an appropriate workload to me.
	SS2	I try my best to accomplish the job assigned by my supervisor.
	SS3	I am willing to continue working under my supervisor regardless of whether or not it will benefit me.
	SS4	I feel easy and comfortable when I communicate with my supervisor.
	SS5	My supervisor credits me for my ideas.
Satisfaction with Colleagues (SC)	SC1 ^a^	I have a good collaboration with my colleagues.
	SC2 ^a^	I am satisfied with the working attitudes of my colleagues.
	SC3 ^a^	My colleagues review themselves to improve their ability to work.
Family Relationship (FR)	FR1	I allot time for my family.
	FR2	My family members show that they love and care for one another.
	FR3 ^a^	My family members do not create any financial burden on me.
Peer Influences (PI)	PI1 ^a^	Good working attitudes of my friends make me feel happy.
	PI2 ^a^	Good quality of life for my friends makes me feel happy.
	PI3	My friends help me when I am having trouble.
Workplace Policies (WP)	WP1	The duration of my annual leave is sufficient.
	WP2	I am satisfied with the activities arranged by the organisation in non-working hours.
	WP3	The organisation allows flexibility in working hours.
	WP4	I am satisfied with the facilities for dependent care provided by the organisation.
	WP5	I am satisfied with the facilities for rest provided by the organisation.
	WP6	I am satisfied with the welfare provided by the organisation.
Job Characteristics (JC)	JC1 ^a^	I do not have to hurry to deal with unexpected tasks.
	JC2	Generally, the work I am tasked with at my organisation is challenging and exciting.
	JC3	I decide when to perform various work tasks.
	JC4	I decide how to perform my work.
Societal Influences (SI)	SI1 ^a^	My colleagues and I are willing to stay in the company after work to continue working.
	SI2 ^a^	I am willing to support the social movement.
	SI3	The government policies developed to address WLB are excellent.
	SI4	I think changing WLB policies now is unnecessary.
	SI5 ^a^	The use of social media makes me feel happy.
Feelings at Work (FW)	FW1	My job is usually interesting enough to keep me from being bored.
	FW2	I feel happy at work.
	FW3 ^a^	Pursuing further studies helps me improve my career development.
Personal Time Use (PTU)	PTU1	I can do what I like to do after work.
	PTU2	I have sufficient time to perform personal duties.
	PTU3 ^a^	I have sufficient time for being solitary.
Health Condition (HC)	HC1	In the last three months, I have had no illnesses.
	HC2	In the last three months, I have had a healthy mental condition.
	HC3	In the last three months, I have had sufficient sleep.
	HC4	In the last three months, I have not woken up in the middle of the night or in the early morning.
	HC5	I get sufficient exercise.
	HC6 ^a^	In the last three months, I eat three square meals per day.
**Subjective Wellbeing**		
Personal Life Satisfaction (PLS)	PLS1	I feel cheerful.
	PLS2	I feel that life is meaningful.
	PLS3	I enjoy my life.
	PLS4	I hold goals and beliefs that affirm a sense of direction in life, and I feel that life has purpose and meaning.
	PLS5	I plan my time for personal matters effectively.
	PLS6	I am satisfied with my personal life.
Health Outcomes (HO)	HO1	In the morning, I wake up and feel ready to get up for the day.
	HO2	In the last three months, I have had no difficulty sleeping.
	HO3	In the past three months, other than my regular job, I have done sufficient physical activities or exercises.
	HO4	In the last three months, I have had healthy and balanced meals.
	HO5	In the last three months, I have never skipped a meal.
	HO6	In the last three months, I have felt confident about my ability to handle my personal problems.
	HO7	In the last three months, I have been able to control irritations in my life.
	HO8	I generally feel happy.
	HO9	My mental health is generally excellent.
	HO10	My physical health is generally excellent.
**Employee Wellbeing**		
Work Attitudes (WA)	WA1	In the last three months, I have been feeling energetic at work.
	WA2	In the last three months, I have been concentrating at work.
	WA3	In the last three months, I have been feeling confident at work.
	WA4	In the last three months, I have been feeling passionate at work.
	WA5	In the last three months, I have been feeling a sense of accomplishment at work.
	WA6	In the last three months, I have been feeling a sense of belonging at work.
	WA7	In the last three months, I have not thought about leaving this job.
	WA8	I am very much personally involved in my work.
	WA9	I am satisfied with my job performance.
Work Behaviours (WB)	WB1	In the last three months, I have been feeling efficient at work.
	WB2 ^a^	In the last three months, I have exceeded my work performance target.
	WB3	I can select the location of where I work.
	WB4	I can schedule when I work (e.g., scheduling hours, time of day).
	WB5	I can schedule what tasks I will do (e.g., content of work, process used).
	WB6 ^a^	My time management ability makes me plan everything effectively.
Career Aspirations (CA)	CA1	The company develops clear career development programmes for employees.
	CA2	My company has programmes and policies that help employees to advance in their functional specialisation.
	CA3	My company has programmes and policies that help employees reach higher managerial levels.
Cooperation with Colleagues (CC)	CC1	My colleagues and I care about each other’s work problems and needs.
	CC2	I feel comfortable with my co-workers.
	CC3	My co-workers and I help each other when facing problems.
**Quality and Quantity of Personal Life-Time**		
Family Context (FC)	FC1 ^a^	I have enough time to communicate with my family (spouse and/or children).
	FC2 ^a^	I have enough time to join in family activities.
	FC3	I have a good relationship with my family.
Leisure Time (LT)	LT1 ^a^	I have enough leisure time.
	LT2 ^a^	I make the best use of my leisure time.
	LT3 ^a^	I have enjoyable leisure time.
Social Life (SL)	SL1	I have enough time for my friends.
	SL2	I am interested in society or social life.
	SL3	I have a sense of belongingness to a community and receive comfort and support from the community.
**Work–life Balance**		
Work–life Balance (WLB)	WLB1	I currently have a good balance between the time I spend at work and the time I have for non-work activities.
	WLB2	I have no difficulty in balancing my work and non-work activities.
	WLB3	I feel that the balance between my work demands and non-work activities is currently appropriate.
	WLB4	Overall, I believe that my work and non-work life are balanced.

Note: ^a^ New survey items.

**Table 2 ijerph-18-04525-t002:** Exploratory factor analysis of the new instruments.

Items	Factor Loading					Communality
	Relational Factors	Community and Societal Factors	Intrapersonal-Level Factors	Employee Wellbeing	Quality and Quantity of Personal Life-Time	
SC1	0.427					0.513
SC2	0.602					0.581
SC3	0.498					0.477
FR3	0.576					0.582
PI1	0.648					0.556
PI2	0.702					0.574
JC1		0.564				0.625
SI1		0.624				0.594
SI2		0.568				0.532
SI5		0.464				0.581
FW3			0.504			0.530
PTU3			0.585			0.579
HC6			0.559			0.561
WB2				0.431		0.583
WB6				0.546		0.651
FC1					0.570	0.440
FC2					0.551	0.560
LT1					0.472	0.501
LT2					0.608	0.507
LT5					0.703	0.640

Abbreviation: SC, Satisfaction with colleagues; FR, Family relationship; PI, Peer influences; JC, Job characteristics; SI, Societal influences; FW, Feelings at work; PTU, Personal time use; HC, Health condition; WB, Work behaviours; FC, Family context; LT, Leisure time.

**Table 3 ijerph-18-04525-t003:** Reliability and convergent validity of the measurement model.

Construct	Item	Factor Loading	Composite Reliability	Average Variance Extracted
Satisfaction with Supervisor	SS1	0.796	0.894	0.629
	SS2	0.795		
	SS3	0.745		
	SS4	0.832		
	SS5	0.795		
Satisfaction with Colleagues	SC1 ^a^	0.728	0.825	0.612
	SC2 ^a^	0.811		
	SC3 ^a^	0.805		
Family Relationship	FR1	0.812	0.808	0.584
	FR2	0.787		
	FR3 ^a^	0.689		
Peer Influences	PI1 ^a^	0.777	0.85	0.653
	PI2 ^a^	0.823		
	PI3	0.824		
Workplace Policies	WP1	0.77	0.888	0.57
	WP2	0.72		
	WP3	0.69		
	WP4	0.821		
	WP5	0.739		
	WP6	0.784		
Job Characteristics	JC1 ^a^	0.765	0.859	0.609
	JC2	0.811		
	JC3	0.797		
	JC4	0.733		
Societal Influences	SI1 ^a^	0.809	0.866	0.565
	SI2 ^a^	0.724		
	SI3	0.757		
	SI4	0.796		
	SI5 ^a^	0.664		
Feelings at Work	FW1	0.736	0.761	0.518
	FW2	0.807		
	FW3 ^a^	0.603		
Personal Time Use	PTU1	0.715	0.814	0.594
	PTU2	0.758		
	PTU3 ^a^	0.835		
Work–Life Balance	WLB1	0.783	0.861	0.609
	WLB2	0.78		
	WLB3	0.825		
	WLB4	0.73		
Health Condition	HC1	0.763	0.878	0.546
	HC2	0.706		
	HC3	0.735		
	HC4	0.767		
	HC5	0.693		
	HC6 ^a^	0.769		
Personal Life Satisfaction	PLS1	0.808	0.906	0.616
	PLS2	0.744		
	PLS3	0.814		
	PLS4	0.789		
	PLS5	0.78		
	PLS6	0.773		
Health Outcomes	HO1	0.786	0.944	0.629
	HO2	0.847		
	HO3	0.868		
	HO4	0.741		
	HO5	0.801		
	HO6	0.801		
	HO7	0.763		
	HO8	0.847		
	HO9	0.777		
	HO10	0.682		
Work Attitudes	WA1	0.799	0.938	0.626
	WA2	0.773		
	WA3	0.782		
	WA4	0.844		
	WA5	0.789		
	WA6	0.791		
	WA7	0.784		
	WA8	0.783		
	WA9	0.776		
Work Behaviours	WB1	0.807	0.919	0.653
	WB2 ^a^	0.786		
	WB3	0.758		
	WB4	0.827		
	WB5	0.818		
	WB6 ^a^	0.85		
Career Aspirations	CA1	0.779	0.847	0.65
	CA2	0.847		
	CA3	0.792		
Cooperation with Colleagues	CC1	0.871	0.858	0.669
	CC2	0.765		
	CC3	0.815		
Family Context	FC1 ^a^	0.685	0.778	0.539
	FC2 ^a^	0.741		
	FC3	0.774		
Leisure Time	LT1 ^a^	0.687	0.752	0.504
	LT2 ^a^	0.65		
	LT3 ^a^	0.786		
Social Life	SI1 ^a^	0.809	0.866	0.565
	SI2 ^a^	0.724		
	SI3	0.757		
	SI4	0.796		
	SI5 ^a^	0.664		

^a^ New survey items.

**Table 4 ijerph-18-04525-t004:** Inter-factor correlations amongst latent variables.

Factor	1	2	3	4	5	6	7
1. Relational factors	0.815						
2. Community and societal factors	0.801 ***	0.829					
3. Intrapersonal-level factors	0.751 ***	0.734 ***	0.857				
4. Work–life balance	0.737 ***	0.746 ***	0.811 ***	0.897			
5. Subjective wellbeing	0.776 ***	0.772 ***	0.804 ***	0.745 ***	0.912		
6. Employee wellbeing	0.787 ***	0.737 ***	0.802 ***	0.799 ***	0.769 ***	0.875	
7. Quality and quantity of personal life-time	0.717 ***	0.768 ***	0.806 ***	0.798 ***	0.754 ***	0.798 ***	0.831

*** *p* ≤ 0.001.

**Table 5 ijerph-18-04525-t005:** Pearson correlation.

Variables	1	2	3	4	5	6	7	8	9	10	11	12	13	14	15	16	17
Gender ^a^	1.000																
Age	−0.269 **	1.000															
Educational level	0.247 **	−0.576 **	1.000														
Marital status ^b^	−0.248 **	0.802 **	−0.543 **	1.000													
Number of children	−0.060 *	0.178 **	−0.019	0.408 **	1.000												
Number of dependents	−0.013	−0.029	0.115 **	0.032	0.252 **	1.000											
Religious belief ^c^	−0.099 **	0.086 **	−0.188 **	0.073 *	−0.048	−0.037	1.000										
Average monthly income	−0.056	0.069 *	0.235 **	0.106 **	0.204 **	0.179 **	−0.035	1.000									
Average working hours per week	−0.163 **	0.390 **	−0.307 **	0.310 **	−0.014	−0.042	0.087 **	0.032	1.000								
Working years	−0.260 **	0.920 **	−0.650 **	0.790 **	0.131 **	−0.065 *	0.120 **	0.018	0.344 **	1.000							
Relational factors	0.031	−0.066 *	0.131 **	−0.082 **	0.001	−0.011	−0.054	0.095 **	−0.071 *	−0.085 **	1.000						
Community and societal factors	−0.041	0.047	−0.009	0.040	0.032	0.018	−0.030	0.165 **	−0.026	0.049	0.801 **	1.000					
Intrapersonal level factors	−0.056	0.030	0.016	0.017	0.038	−0.007	−0.073 *	0.134 **	−0.098 **	0.023	0.851 **	0.834 **	1.000				
Subjective wellbeing	−0.061 *	0.051	0.004	0.057	0.070 *	0.009	−0.042	0.135 **	−0.056	0.051	0.776 **	0.772 **	0.854 **	1.000			
Employee wellbeing	−0.042	0.043	0.003	0.034	0.038	0.002	−0.023	0.146 **	−0.033	0.045	0.787 **	0.837 **	0.820 **	0.869 **	1.000		
Quality and quantity of personal life-time	0.009	−0.005	0.068 *	−0.012	0.015	−0.006	−0.027	0.121 **	−0.121 **	−0.017	0.817 **	0.768 **	0.856 **	0.854 **	0.798 **	1.000	
Work–life balance	−0.062 *	0.079 **	−0.010	0.068 *	0.039	−0.020	−0.012	0.117 **	−0.0091 **	0.075 *	0.737 **	0.746 **	0.819 **	0.845 **	0.799 **	0.798 **	1.000

^a^ 0 = Male; 1 = Female. ^b^ 0 = Single; 1 = Single with children; 2 = Married and without children; 3 = Married and with child/children; 4 = Divorced/separated/widowed. ^c^ 0 = Christianity; 1 = Buddhism; 2 = Islam; 3 = No; 4 = Other. * *p* ≤ 0.05. ** *p* ≤ 0.01.

**Table 6 ijerph-18-04525-t006:** Regression of moderator effect of subjective wellbeing on the relationship between WLB with respect to employee wellbeing and quality and quantity of personal life-time.

Dependent Variables	Independent Variables	b	SE	*t*
Employee wellbeing	(Constant)	1.048	0.152	6.903
	WLB	0.115 **	0.042	2.733
	Subjective wellbeing	0.566 ***	0.044	12.991
	WLB X Subjective wellbeing	0.016 **	0.008	1.948
Quality and quantity of personal life-time	(Constant)	0.593	0.162	3.657
	WLB	0.324 ***	0.045	7.193
	Subjective wellbeing	0.701 ***	0.046	15.077
	WLB X Subjective wellbeing	0.023 **	0.009	−2.597

** *p* ≤ 0.01. *** *p* ≤ 0.001.

## Data Availability

Data not available due to ethical restrictions.

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
