# Peer review of "The Interplay of Socioecological Determinants of Work–Life Balance, Subjective Wellbeing and Employee Wellbeing"

_ijerph, 2021, doi:10.3390/ijerph18094525_

Round 1
Reviewer 1 Report
The paper is well written and organized. The reviewer appreciates the challenges of gathering information from surveys, interpreting the data and providing sound conclusions. It follows some publications on the influence of the environment on wellbeing (e.g. nature/green trees). A few suggestions:
line 61- "life or the counter-off of between"- I think the word "of" can be deleted.
Table 5-Under variables- I think it should read- Gendera - I do not think is Gendera.
Figure 1- All the numbers have *** and then the explanation about p. Suggestion- Add the explanation to the title : Figure 1. Result of the structural mode with standardised estimates where p ≤ 0.001.
Lines 460-461- As future research is planned, it may be good to understand the impact of personality types to the overall wellbeing picture. Those lines hinted that there may be some relationship between "optimistic" people and wellbeing versus "pesimistic" people.
Line 479- "In a nutshell, this study advances the WLB literature by offering a profound perspective..."- instead of a profound perspective, what about novel approach or new perspective?
Future consideration- there is a lot of data on the paper. Maybe future papers can include additional analyses to understand gender differences, educational levels, social differences, etc.
Overall, the paper is well done.
Author Response
Response to Reviewer 1 Comments
The paper is well written and organized. The reviewer appreciates the challenges of gathering information from surveys, interpreting the data and providing sound conclusions. It follows some publications on the influence of the environment on wellbeing (e.g. nature/green trees). A few suggestions:
Point 1: line 61- "life or the counter-off of between"- I think the word "of" can be deleted.
Response 1: Thank you very much for pointing the error. We have deleted “of”.
Point 2: Table 5-Under variables- I think it should read- Gendera - I do not think is Gendera.
Response 2: Thanks for highlighting this. We have changed Gendera to Gendera.
Point 3: Figure 1- All the numbers have *** and then the explanation about p. Suggestion- Add the explanation to the title : Figure 1. Result of the structural mode with standardised estimates where p ≤ 0.001.
Response 3: Thank you for your suggestion. “where p ≤ 0.001” was added to the title of Figure 1. See line 601.
Point 4: Lines 460-461- As future research is planned, it may be good to understand the impact of personality types to the overall wellbeing picture. Those lines hinted that there may be some relationship between "optimistic" people and wellbeing versus "pesimistic" people.
Response 4: Thank you for your suggestion. A research recommendation, “In addition, given the predominant role of intrapersonal factors on WLB and wellbeing, the impact of personality (e.g. pessimistic, optimistic and envious) on wellbeing can be assessed and different personality types on wellbeing can be identified.” was added in lines 787-789.
Point 5: Line 479- "In a nutshell, this study advances the WLB literature by offering a profound perspective..."- instead of a profound perspective, what about novel approach or new perspective?
Response 5: Thank you for your comment. “Furthermore, we suggested a novel approach, socioecological systems, in studying the determinants of work-life balance that the dynamic interaction between environmental and individual factors was identified.” was added in lines 642-644 to explain the novel approach suggested in this study.
Point 6: Future consideration- there is a lot of data on the paper. Maybe future papers can include additional analyses to understand gender differences, educational levels, social differences, etc.
Response 6: Thank you for your suggestion. We will include them in future research due to the length of paper.
Point 7: Overall, the paper is well done.
Response 7: Thank you again for your comments and suggestions and we hope the revised paper will meet with your approval in our pursuit of publishing in the International Journal of Environmental Research and Public Health.

Reviewer 2 Report
- The paper deals with an interesting topic in “HRM”. I appreciated the opportunity to read your paper. Your story is very interesting.
- There is a lot of information included in the introduction, but it may be better placed in the new 2. "theoretical" section. I would also encourage you to review the paragraph structure of the paper to enhance flow and readability. In the introduction, you should clarify objectives and implications. Is the most important information in this section, but in the manuscript is difficult because you include too much information.
- The manuscript version is explicitly show the contribution of author(s) in existing literature.
- The given literature is up-to-the date. I am satisfied with the quality and quantity of the provided literature, but but, you can review the MDPI article related to the research topic:
https://www.mdpi.com/1660-4601/17/6/1920
- The methodology section is appropriate and clear.
- In this paper, implications in the context of theory, practice, and polices are clearly given. Good luck.
- The quality of communication is clear and acceptable. Good luck for your achievements.
- The paper clearly expresses its case, measured against the technical language of the field and the expected knowledge of the journal's readership. The authors of the article presented the issue discussed in a clear and legible way.
Author Response
Response to Reviewer 2 Comments
Point 1: The paper deals with an interesting topic in “HRM”. I appreciated the opportunity to read your paper. Your story is very interesting.
Response 1: Thank you very much for your time and effort to review our paper. We appreciate your positive assessment of our paper, but more importantly, we are grateful for your constructive comments and suggestions.
Point 2: There is a lot of information included in the introduction, but it may be better placed in the new 2. "theoretical" section. I would also encourage you to review the paragraph structure of the paper to enhance flow and readability. In the introduction, you should clarify objectives and implications. Is the most important information in this section, but in the manuscript is difficult because you include too much information.
Response 2: Thank you very much. To reduce the length in the introduction section, the introduction section was separated with a new section titled “1.1 Literature Review”. Theoretical Background and Hypotheses has been organized as section “1.2 Theoretical Background and Hypotheses”.
The objective of this study is stated in lines 45-50 which is “This study aims to understand the interplay between these factors and WLB beyond the typical focus (e.g. the direct influencing power of factors on WLB) and move the research onward by testing more complex relationships regarding WLB.”
The implication is discussed in lines 133-135 which is “Overall, useful insights into the importance of the influencing power of external and internal factors influencing WLB were provided by this study, and empirical evidence demonstrating the importance of WLB for individuals and organisations was shown.”
Point 3: The manuscript version is explicitly show the contribution of author(s) in existing literature.
Response 3: Thank you very much for your positive comments.
Point 4: The given literature is up-to-the date. I am satisfied with the quality and quantity of the provided literature, but but, you can review the MDPI article related to the research topic:
https://www.mdpi.com/1660-4601/17/6/1920
Response 4: Thank you for your suggestion. This paper is cited in the section practical implications that is “In this study, several social needs were identified: relatedness with supervisor and colleagues, workplace policies, job autonomy and interestingness of job tasks. In other words, organisations can formulate tailor-made workplace policies to fulfil the feasible and reasonable requirements of workers so that workers can flexibly manage the work–life interface [109-112].” in lines 741-745.
Point 5: The methodology section is appropriate and clear.
Response 5: Thank you very much.
Point 6: In this paper, implications in the context of theory, practice, and polices are clearly given. Good luck.
Response 6: Thank you very much.
Point 7: The quality of communication is clear and acceptable. Good luck for your achievements.
Response 7: Thank you very much.
Point 8: The paper clearly expresses its case, measured against the technical language of the field and the expected knowledge of the journal's readership. The authors of the article presented the issue discussed in a clear and legible way.
Response 8: Thank you very much. We hope that you will be happy with the revised paper and that the revised paper will meet with your approval in our pursuit of publishing in the International Journal of Environmental Research and Public Health.

Reviewer 3 Report
Thank you for the opportunity to review the manuscript The Interplay of Socioecological Determinants of Work–Life Balance, Subjective Wellbeing and Employee Wellbeing”.
Congratulations to the authors for their work, I found your paper a potentially very valuable resource on Health Science and therefore an interesting and relevant contribution to IJERPH.
The manuscript explores the socioecological factors that influenced work–life balance (WLB) and how they operated.
However, in my opinion there are several aspects should be revised to improve the explanatory power of the manuscript as noted below.
SPECIFIC COMMENTS:
TITTLE
Correct.
ABSTRACT
In the abstract the results and conclusions seem mixed. It is recommended to structure the information.
INTRODUCTION
The introduction is too long and sometimes confusing, as the authors mix it up with the study hypotheses. Please structure the information and put the hypotheses of the study after exposing the most relevant theoretical concepts of the research.
Where are the objectives of the study?
METHOD AND RESULTS
- The characteristics of the sample must be in the results section.
- What was the sample universe? Was the sample representative?
- What were the inclusion / exclusion criteria?
- The elaboration of the questionnaire used is confusing and the information is not structured. Please put the information in an orderly manner (bibliography search, translation / back translation, etc.).
- Does the questionnaire prepared by the authors have validity of content? Justify.
- The data collection procedure is poorly explained. Please explain in more detail.
- The method lacks a section on ethical considerations and data analysis.
- The first time that the authors describe that they are going to perform an exploratory / confirmatory factor analysis is in the results section.
- In the exploratory factor analysis, what models were tested? What method for its analysis was used?
- The authors say that the CFA model fits well, however they report RMSEA = 0.075. This would indicate a poor fit of the model. Please justify.
- In the Preliminary analyzes of the measurement model section, the authors mix information that should be in the method section (... adopting SPSS 24.0 and AMOS 24.0.).
- What parameter estimation method have you used for the CFA [Maximum likelihood (ML), unweighted least squares (ULS) and diagonally weighted least squares (DWLS)]? Please justify.
- To see the final model, please include the path diagram with the standardized parameter estimation. This could replace the information in Tables 3 and 4.
- The model proposed in section 3.2. Test of hypotheses does not fit (RMSEA = 0.080). All the assertions that are made could be false due to the lack of fit of the model. What method was used to estimate the parameters [Maximum likelihood (ML), unweighted least squares (ULS) and diagonally weighted least squares (DWLS)]?
DISCUSSION AND CONCLUSION
Correct
Author Response
Response to Reviewer 3 Comments
Point 1: Thank you for the opportunity to review the manuscript The Interplay of Socioecological Determinants of Work–Life Balance, Subjective Wellbeing and Employee Wellbeing”.
Congratulations to the authors for their work, I found your paper a potentially very valuable resource on Health Science and therefore an interesting and relevant contribution to IJERPH.
The manuscript explores the socioecological factors that influenced work–life balance (WLB) and how they operated.
However, in my opinion there are several aspects should be revised to improve the explanatory power of the manuscript as noted below.
Response 1: Thank you for your positive evaluation of our paper. We really appreciate it.
We would also like to thank you for your time and effort to review our paper.
We have taken on board your constructive comments and suggestions, which we have attempted to address objectively, systematically, and diligently.
SPECIFIC COMMENTS:
Point 2: TITTLE
Correct.
Response 2: Thank you very much.
ABSTRACT
Point 3: In the abstract the results and conclusions seem mixed. It is recommended to structure the information.
Response 3: Thank you for highlighting this point.
In the abstract, the results are discussed at lines 15-20 - “The results show that relational, community and societal factors directly influenced the individual factors and were indirectly associated with perceived WLB. Individual factors (i.e. personal feelings, behaviours and health) were found to be the crucial determinants of an individual’s perceived WLB. It was found that WLB positively correlated with employee wellbeing and quality and quantity of personal life-time. Subjective wellbeing was found to be a significant moderator in the relationship between WLB and its outcomes.”
The conclusion was explained at lines 21-23 - “This study demonstrates the process of how workers determine their own WLB by applying the socioecological framework for categorising the determinants, and suggests new avenues that improve the whole wellbeing of workers and also foster long-term development of organisations.”
INTRODUCTION
Point 4: The introduction is too long and sometimes confusing, as the authors mix it up with the study hypotheses. Please structure the information and put the hypotheses of the study after exposing the most relevant theoretical concepts of the research.
Response 4: Thank you very much. To reduce the length in the introduction section, the introduction section was separated with a new section titled “1.1 Literature Review”. Theoretical Background and Hypotheses has been organized as section “1.2 Theoretical Background and Hypotheses”.
Point 5: Where are the objectives of the study?
Response 5: Thank you for raining this point. The objective of this study is stated in lines 45-50 which is “This study aims to understand the interplay between these factors and WLB beyond the typical focus (e.g. the direct influencing power of factors on WLB) and move the research onward by testing more complex relationships regarding WLB.”
METHOD AND RESULTS
Point 6: The characteristics of the sample must be in the results section.
Response 6: Thank you for your suggestion. A new section titled “3.1. Characteristics of the Sample”, is added in results section in lines 494-512. The characteristics of the sample mentioned in the section of Materials and Method was deleted.
Point 7: What was the sample universe? Was the sample representative?
Response 7: Thank you for raising this concern. We wish to gently clarify that our sample selection is explained in lines 472-480.
“Based on simple random sampling, invitation emails were sent to twenty-five trade unions according to the list from Hong Kong Confederation of Trade Unions (HKCTU) [89]. HKCTU is an influential labours group in Hong Kong and several researchers have collected sample data based on the affiliates under this union [90,91]. Finally, three trade unions assisted in distributing the online survey and nine organisations participated in conducting the survey. The sample comprised the workers of key industries of Hong Kong, namely financial services, tourism, trading and logistics, professional services, producer services, cultural and creative industries, medical services and innovation and technology.”
We hope that these efforts help to clarify your concerns on the sample representative in our study.
Point 8: What were the inclusion / exclusion criteria?
Response 8: Thank you very much. “The inclusion criteria were to be aged eighteen and above and currently employed in Hong Kong.” were added in lines 471-472.
Point 9: The elaboration of the questionnaire used is confusing and the information is not structured. Please put the information in an orderly manner (bibliography search, translation / back translation, etc.).
Response 9: Thank you. “The language used for the questionnaire was Chinese. Four independent bilingual translators, who were not associated with the study, were invited to conduct the translation to ensure the reliability of the translated questionnaire [58]. Two of the translators translated the English into Chinese version, and another translated the Chinese into English version. The discrepancies amongst the translators were identified and discussed with the researchers until all were satisfied with the statements.” was shown in lines 308-313 to indicate the translation process.
Point 10: Does the questionnaire prepared by the authors have validity of content? Justify.
Response 10: Thank you very much. “A total of 91 statements regarding the perceived determinants and outcomes of WLB were generated, based on the review of previous studies, and twenty of these were new survey items. The other 71 items were validated by researchers (e.g. [58-60]).” was added in lines 303-306 to show the validity of the content.
Point 11: The data collection procedure is poorly explained. Please explain in more detail.
Response 11: Thank you very much. The discussion on data collection procedure has been expanded.
“The data were collected from an online survey platform, Qualtrics. The procedure of the online survey was approved by the College Research Ethics Sub-committee of City University of Hong Kong. This survey was a self-report design that was administrated online to gather confidential data from participants in different organisations. The first page of the survey explained the background of the survey and the participant agreement. If the individuals understood and consented to the conditions, they could click ‘Agree’ to start the survey. The inclusion criteria were to be aged eighteen and above and currently employed in Hong Kong. Based on simple random sampling, invitation emails were sent to a number of trade unions and organisations. Finally, three trade unions assisted in distributing the online survey and nine organisations participated in conducting the survey.” was demonstrated in lines 465-480.
Point 12: The method lacks a section on ethical considerations and data analysis.
Response 12: Thank you very much. We have added the research ethics details - “The procedure of online survey was approved by College Research Ethics Sub-committee of City University of Hong Kong.” in lines 465-467.
Point 13: The first time that the authors describe that they are going to perform an exploratory/ confirmatory factor analysis is in the results section.
Response 13: Thank you very much. A new section, 2.3. Data Analysis, was added in Methods (lines 482-493) to describe exploratory / confirmatory factor analysis. The contents are as follow.
Exploratory factor analysis (EFA) was adopted to test the preliminary factor analysis of the instruments by using SPSS 24.0. The measurement and structural model were analysed through confirmatory factor analysis (CFA) and SEM, respectively, using SPSS 24.0 and AMOS 24.0. The factor loadings in EFA should be greater than 0.4 to meet the standard [92]. Maximum likelihood estimation (MLE) was used for the models of CFA and SEM. To assess the fitness of the measurement and structural model, five goodness-of fit indices, namely, chi-square to its degree of freedom (X2/df), Tucker–Lewis Index (TLI), comparative fit index (CFI), standardized root mean squared residual (SRMR) and root mean square error of approximation (RMSEA) [93]. Factor loadings, composite reliability (CR) and average variance extracted (AVE) were used for convergent validity. The square root of AVE for the latent factors was larger than the inter-factor correlations indicated the verification of discriminant validity [94].
Point 14: In the exploratory factor analysis, what models were tested? What method for its analysis was used?
Response 14: Thank you very much. In the exploratory factor analysis, the preliminary factor analysis of the new instruments was tested. Thus, “Exploratory factor analysis (EFA) was adopted to test the preliminary factor analysis of the instruments by using SPSS 24.0.” was added in lines 482-483.
Point 15: The authors say that the CFA model fits well, however they report RMSEA = 0.075. This would indicate a poor fit of the model. Please justify.
Response 15: Thank you very much. “A RMSEA value up to 0.08 represents reasonable errors of approximation in the population [96,97].” was added in lines 521-522 to justify the value of RMSEA of the CFA model fitness.
Point 16: In the Preliminary analyzes of the measurement model section, the authors mix information that should be in the method section (... adopting SPSS 24.0 and AMOS 24.0.).
Response 16: Thank you for your suggestions. “Exploratory factor analysis (EFA) was adopted to test the preliminary factor analysis of the instruments by using SPSS 24.0. The measurement and structural model were analysed through confirmatory factor analysis (CFA) and SEM, respectively, using SPSS 24.0 and AMOS 24.0” was added to the section of methods-2.3 Data Analysis (lines 482-485).
Point 17: What parameter estimation method have you used for the CFA [Maximum likelihood (ML), unweighted least squares (ULS) and diagonally weighted least squares (DWLS)]? Please justify.
Response 17: Thank you very much. We used maximum likelihood estimation (MLE) for the model of CFA and Chi-square to its degree of freedom (X2/df), Tucker–Lewis Index (TLI), comparative fit index (CFI), standardized root mean squared residual (SRMR), and root mean square error of approximation (RMSEA) were used for testing the fitness of the CFA model. “Maximum likelihood estimation (MLE) was used for the models of CFA and SEM.” was added in line 486-487. “The results of CFA showed an acceptable overall model fit, except χ2/df (χ2/df = 10.48, CFI = 0.925, TLI = 0.909, RMSEA = 0.075, SRMR = 0.058; [93]).” was shown in lines 516-518.
Point 18: To see the final model, please include the path diagram with the standardized parameter estimation. This could replace the information in Tables 3 and 4.
Response 18: Thank you very much. Figure 1 show the standardized parameter estimation of the structural model which was in page 16.
Point 19: The model proposed in section 3.2. Test of hypotheses does not fit (RMSEA = 0.080). All the assertions that are made could be false due to the lack of fit of the model. What method was used to estimate the parameters [Maximum likelihood (ML), unweighted least squares (ULS) and diagonally weighted least squares (DWLS)]?
Response 19: Thank you very much. To justify the value of RMSEA, “A RMSEA value up to 0.08 represents reasonable errors of approximation in the population [96,97].” was inserted in lines 562-563.
Maximum likelihood estimation (MLE) for the model of SEM was used and the model of SEM was tested by Chi-square to its degree of freedom (X2/df), Tucker–Lewis Index (TLI), comparative fit index (CFI), standardized root mean squared residual (SRMR), and root mean square error of approximation (RMSEA). Therefore, “Maximum likelihood estimation (MLE) was used for the models of CFA and SEM.” was added in line 486-487. “To verify the structural model, SEM was used, and the overall model satisfied the criteria of the indexes of goodness-of-fitness, except χ2/df (χ2/df = 9.31, CFI = 0.933, TLI = 0.920, RMSEA = 0.080, SRMR = 0.059; [93]).” was demonstrated in lines 560-562.
Figure 1 showed the standardised estimates of SEM.
Point 20: DISCUSSION AND CONCLUSION
Correct
Response 20: Thank you very much again for your constructive comments and suggestions, which we have used to improve the quality of our paper. We genuinely felt they were very useful. We hope that the revised paper will now meet with your approval in our pursuit of publishing in the International Journal of Environmental Research and Public Health.

Round 2
Reviewer 3 Report
The authors have addressed all comments.